# Novel TiO_2_/GO-Al_2_O_3_ Hollow Fiber Nanofiltration Membrane for Desalination and Lignin Recovery

**DOI:** 10.3390/membranes12100950

**Published:** 2022-09-28

**Authors:** Xuelong Zhuang, Edoardo Magnone, Min Chang Shin, Jeong In Lee, Jae Yeon Hwang, Young Chan Choi, Jung Hoon Park

**Affiliations:** 1Department of Chemical and Biochemical Engineering, Dongguk University, 30, Pildong-ro 1 gil, Jung-gu, Seoul 04620, Korea; 2Fine Dust Research, Korea Institute of Energy Research (KIER), Daejeon 34129, Korea

**Keywords:** nanofiltration, TiO_2_, graphene oxide (GO), Al_2_O_3_ hollow fiber, sodium ion removal, lignin recycling

## Abstract

Due to its greater physical–chemical stability, ceramic nanofiltration (NF) membranes were used in a number of industrial applications. In this study, a novel NF membrane was prepared by co-depositing a titanium dioxide (TiO_2_) and graphene oxide (GO) composite layer directly onto a porous α-Al_2_O_3_ hollow fiber (HF) support. An 8 µm-thick TiO_2_/GO layer was deposited to the surface of α-Al2O3 HF support by vacuum deposition method to produce advanced TiO_2_/GO-Al_2_O_3_ HF NF membrane. Scanning electron microscope (SEM) micrographs, energy dispersive spectrometer (EDS), X-ray powder diffraction (XRD), thermogravimetric analyzer (TGA), porosity, 3-point bending strength, zeta potential analysis, and hydrophilic properties by water contact angle are used for TiO_2_/GO-Al_2_O_3_ HF NF membrane characterization. The results show that the developed membrane’s MWCO ranged from 600 to 800 Da. The water flux, rejection of lignin, and sodium ions were 5.6 L/m^2^ h·bar, ~92.1%, and ~5.5%, respectively. In a five-day NF process, the TiO_2_/GO-Al_2_O_3_ HF NF membrane exhibits good lignin permeation stability of about 14.5 L/m^2^ h.

## 1. Introduction

It is well known that wood chips are utilized as the primary raw material in the papermaking process in order to make the paper required for daily use. The presence of lignin—the third major component in lignocellulosic biomass—in the cellulose pulp mixture causes the yellowing of the paper and its durability [1]. Thus, to avoid this phenomenon, a NaOH solution is added to the paper-making process to dissolve and then separate the lignin from cellulose [2]. The primary component of the black liquor, the separated lignin, is usually discharged as wastewater. Large amounts of wastewater containing lignin, an abundant natural resource with an aromatic structure, are discharged annually [3], which can be used as a polymeric material for the synthesis of high-value chemicals and carbon-based products [4]. Given that the wastewater also has high levels of residual Na^+^ ions, using lignin directly for other applications is challenging. In order to recover more pure lignin for its reutilization, it is now a prominent research issue to remove Na^+^ ions from wastewater [5] and, to overcome this problem, it is crucial to find alternative solutions. Despite being challenging, recovering lignin in the presence of Na^+^ ions is a practicable strategy. There has been some progress recently [6,7,8,9,10]. Most earlier investigations have employed polymeric membranes to separate the remaining lignin from the effluent black liquors via the nanofiltration (NF) process [6,7,8,9,11]. The membrane pore sizes used in NF, a pressure-driven membrane separation process, are bigger than those used in reverse osmosis membranes. The NF membrane has a molecular weight cut-off (MWCO) of less than 1000 Da, and its pore sizes typically range from 1 to 2 nanometers. Since lignin has a molecular weight that is between 0.5 and 10 kDa, NF membranes with effective pore sizes between 1 and 2 nm are required [4,5,6,7,8,9,10,11].

Conventional polymeric membranes are often limited by weak chemical stability, while they can offer good lignin rejections. Compared to the polymeric membranes, the ceramic membrane displayed good stability but lower lignin retention [12].

Due to their superior physical, thermal, and chemical durability to organic solvents and acid-base corrosion, inorganic NF membranes are considered an advanced substitute for polymeric membranes [13]. α-Al_2_O_3_ has attracted a lot of interest in the context of the NF process as support for various other oxides [14,15,16,17,18,19,20,21,22,23,24,25]. In this area of interest, the top layers of α-Al_2_O_3_ supports for separation applications are frequently composed of γ-Al_2_O_3_ [17,18,19], TiO_2_ [20,21,22], GO [23,24], and so on [25]. In detail, in our previous study, we found that the ultra-thin γ-Al_2_O_3_ film-coated porous α-Al_2_O_3_ hollow fiber (HF) support can be applied with success for the purification and concentration of lignin in alkaline media [17].

Sotto et al. [20] prepared polyethersulfone (PESf) membranes by the addition of TiO_2_ nanoparticles during membrane synthesis to increase the permeability and fouling resistance. Chen et al. [21] successfully prepared a disk-shaped TiO_2_/α-Al_2_O_3_ NF membrane and they found that the retention rates for MgSO_4_, MgCl_2_, Na_2_SO_4_, and NaCl solutions were 77.8%, 74.1%, 53.7%, and 24.7%, respectively. The disk-shaped TiO_2_/α-Al_2_O_3_ NF membrane has a permeate flow of approximately 2.53 L/m^2^ h bar [21]. Khalili et al. [22] investigated the rejection of the chloride ion using γ-Al_2_O_3_/TiO_2_ composite membranes deposited on α-Al_2_O_3_ support discs. The chloride ion rejection varies from about 60% to 85% at different pH values and various concentrations of NaCl solution [22]. In addition, the TiO_2_ NF membrane has greater chemical stability than Al_2_O_3_ does [26].

Due to its unique structural properties, great mechanical strength and low cost, graphene oxide (GO) has received a lot of attention in recent years from various researchers as a possible membrane material for use in water purification systems [7]. Wang et al. [23] did research on the preparation of a GO-Al_2_O_3_ NF tubular membrane for the desalination process with a Na_2_SO_4_ rejection of 91%. The order of NF membranes’ rejections toward four salt solutions was Na_2_SO_4_ > MgSO_4_ > NaCl > MgCl_2_ [23]. Hu et al. [24] prepared a crack-free disk-shaped GO-Al_2_O_3_ membrane with a GO thickness of 800 nm. In their experiments, the salt rejection of the membrane reaches 28.66%, 39.24% and 43.52% for NaCl, Cu(NO_3_)_2_ and MgSO_4_, respectively [24]. In addition, TiO_2_ nanoparticles were used as intercalators by Xu et al. to improve GO performance [27].

The original idea behind the present work is to study the potential synergic combination between TiO_2_ and GO through an inorganic NF process based on the novel TiO_2_/GO-Al_2_O_3_ HF NF membrane. To the best of our knowledge, this is the first study that considers depositing a composite layer of TiO_2_ and GO directly onto a porous α-Al_2_O_3_ HF support for desalination and lignin recovery from an alkaline lignin aqueous solution. The findings show that the proposed TiO_2_/GO-Al_2_O_3_ HF NF membrane has an efficient salt rejection toward sodium sulfate (Na_2_SO_4_), magnesium sulfate (MgSO_4_), sodium chloride (NaCl), potassium chloride (KCl), calcium chloride (CaCl_2_), magnesium chloride (MgCl_2_), and aluminum chloride (AlCl_3_).

An excellent lignin retention rate of more than 92% is achieved with the TiO_2_/GO-Al_2_O_3_ HF NF membrane.

## 2. Experimental Section

### 2.1. Materials

α-Al_2_O_3_ powder (<0.5 μm) and Polyethersulfone (PESf) were purchased from Kceracell (Boksu-myeon, Korea) and Ultrason® (Lemförde, Germany), respectively. Polyvinylpyrrolidone (PVP, 99.5%) and polyvinyl alcohol (PVA, MW = 1800 Da) were both acquired from Sigma (MO, USA). Na_2_SO_4_ (99.0%), MgSO_4_ (98.0%), NaCl (99.0%), KCl (99.0%), CaCl_2_ (99.0%), MgCl_2_ (99.0%), AlCl_3_ (99.0%), 1-Methyl-2-pyrrolidinone anhydrous (NMP, 99.5%), Aluminum isopropoxide (AIP, 98%), Sulfuric acid, (H_2_SO_4_, 70%), Potassium permanganate (KMnO_4_, 99.3%), Hydrogen peroxide (H_2_O_2_, 3.0%), Titanium dioxide (anatase-TiO_2_), Graphite (C, powder), Nitric acid (HNO_3_, 60%), Hydrochloric acid (HCl, 10%) and Polyethylene glycol (PEG) with 200 Da, 400 Da, 600 Da, 800 Da, 1000 Da, 1500 Da, 2000 Da, 4000 Da, 6000 Da molecular weight were obtained from Samchun Chemical Co., Ltd. (Pyeongtaek, Korea) and used without further purification. The lignin wastewater was provided by the Korean Institute of Energy Research (KIER). The pH, total organic carbon (TOC), Na^+^ concentration, and Lignin concentration were 14, 9780 mg/kg, 195 mg/L, and 3456 mg/kg, respectively. All of the water used in this work was deionized.

### 2.2. Preparation of α-Al_2_O_3_ Hollow Fiber Support

Porous α-Al_2_O_3_ supports were successfully obtained using our prior knowledge in the production of α-Al_2_O_3_ hollow fiber membranes [17,28]. In brief, 36 g of PESf was added to 201 g of NMP and stirred at 150 rpm for one day to obtain a homogenous solution. After one day, 3 g of PVP and 360 g of α-Al_2_O_3_ powder were added to the solution and then stirred again (300 rpm) for another day. The prepared casting solution was placed in a stainless steel reactor and was defoamed for an hour at 0.8 bar under vacuum. The air gap distance was set at 10 cm and the nitrogen pressure used was 3 Mpa. The water flow rate was 10 mL/min. After spinning, the produced green body was immersed in water for one day to remove any remaining organic solvents and then dried for an entire night at 90 °C in a static oven to eliminate any remaining moisture. The green body was thermally treated at high temperature in inert atmosphere (1300 °C, 3 h).

### 2.3. Preparation of Graphene Oxide (GO)

The Hummers method was used to produce the GO powder by chemically exfoliating graphite with an oxidant [29,30]. H_2_SO_4_ was used as an oxidant. In a dried three-neck flask, 100 mL of concentrated H_2_SO_4_ that had been previously chilled in an ice-water bath was added. The reaction was carried out by progressively adding 12.5 g of KMnO_4_ and 0.625 g of C powder while stirring at 400 rpm in the ice-water bath. Following this procedure, the three-neck flask was placed in a water bath at a constant temperature of 35 °C for three hours. The solution was then gradually transferred to a beaker containing 585 mL of deionized water after cooling to room temperature. Then, 35 mL of HCl and 17 mL of H_2_O_2_ were added instantly and the mixture was vigorously agitated until a bright yellow reaction product was obtained. The resultant reaction product was then repeatedly alternatively washed with HCl and deionized water to remove the soluble ions. The resulting precipitate was then centrifuged after being treated with deionized water. To achieve the final GO powder, the centrifuged precipitate was dried for two days.

### 2.4. Preparation of TiO_2_/GO-Al_2_O_3_ Hollow Fiber (HF) NANOFILTRATION (NF) Membrane

A vacuum-assisted technique was used to co-deposit TiO_2_ and GO powders on the Al_2_O_3_ HF’s outer surface. To make GO suspension, 0.2 g of GO powder was dissolved in one liter of water and placed in an ultrasonic vibration unit for 15 min. To make TiO_2_ suspension, we follow the same GO methodology. To prepare the co-coating solution, TiO_2_ and GO suspensions were then properly mixed by an ultrasonic process in a 1:1 ratio for 15 min. Teflon tape is used to seal one end of the Al_2_O_3_ HF support, while the other end is attached to a vacuum pump (1 bar). After that, the Al_2_O_3_ HF support is immersed in the TiO_2_ and GO coating solution for 15 min. After the TiO_2_ and GO coating process, the Al_2_O_3_ HF support was placed in air and vacuumed for another 15 min at the same vacuum pressure. This procedure provided a guarantee that TiO_2_ and GO coating adhered effectively to the Al_2_O_3_ HF support. The membrane was put in an oven at 70 °C for one night and then washed in water as the final phase of the procedure to prepare the TiO_2_/GO-Al_2_O_3_ HF NF membranes.

### 2.5. Characterization of Al_2_O_3_ HF Support and TiO_2_/GO-Al_2_O_3_ HF NF Membrane

A scanning electron microscope (SEM, NovaNano SEM450/FEI, OA, USA) and energy dispersive spectrometer (EDS, NNS-450/FEI, OA, USA) were used to analyze the surface structure of the Al_2_O_3_ HF support and TiO_2_/GO-Al_2_O_3_ HF NF membrane to confirm the TiO_2_/GO co-deposition. In addition, a 15.00 kV electron beam was used to evaluate the coating’s thickness and establish the integrity of the TiO_2_ and GO coating layer. To establish the effective presence of the coating’s Ti, C, and other elements, EDS analysis was also performed. XRD analysis (Cu Kα1, λ = 1.54041 Å, Dmax-2500pc, Rigaku, Japan) was performed to confirm the material composition. A Thermogravimetric Analyzer (TGA, SDT-Q600/TA, DE, USA) was used to measure the thermal stability of the Al_2_O_3_ HF support and TiO_2_/GO-Al_2_O_3_ HF NF membrane at a temperature of 800 °C in high-quality air gas. The fixed gas flow rate was 20 mL/min. A UV–Vis instrument used was a Cary 100 Conc/VARIAN (CA, USA).

The membranes’ porosity *ε* (%) was calculated using the weight difference method [17]. The sample was immersed in distilled water for two days as the initial stage. Next, the water in the sample cavity was blown out with nitrogen gas and the residual humidity was dried on the sample’s surface. The sample was then weighed and the result was noted as *m_wet_*. In the second step, the same sample was heated to 105 °C for two days, weighed, and its weight was recorded as *m_dry_*. The porosity was then determined using the following formula:(1)$$\varepsilon =\frac{∆m}{V\rho_{H_{2}O}}\times 100\%=\frac{m_{wet}-m_{dry}}{\frac{\pi }{4}\left(d_{o}^{2}-d_{i}^{2}\right)l\rho_{H_{2}O}}\times 100\%\;$$
where *d_o_*, *d_i_ l*, and *ρ_H_*_2*O*_ are the outer diameter (cm) and inner diameter (cm), length (cm), and water density (1.0 g/cm^3^) of the membrane, respectively.

By applying a 0.1 mL drop of distilled water to the sample surface, a water contact angle analyzer (SEO Phoenix-I, Korea) was used to measure the water contact angle of the Al_2_O_3_ HF support and TiO_2_/GO-Al_2_O_3_ HF NF membrane. In total, 500 ms intervals were used to capture the drop in images. These findings were utilized to examine the sample’s ability to absorb water as well as its hydrophilic characteristics.

A three-point bending test using a Universal Testing Machine (Kyoungsung Testing Machine, Ansan-si, Korea) was used to determine the membrane’s mechanical strength. The drop speed was set to 1 mm/min and the distance between the two points was adjusted to 1 cm. The following formula was used to determine the mechanical strength F (Mpa) results:(2)F=8fLD0πD04−Di4 
where *f*, *L*, *D*_0_ and *D_i_* are the measured load at which fracture occurs (Newton), the length (meter), the outer diameter (meter), and the inner diameter (meter) of the membranes, respectively.

Additionally, zeta potential analysis was conducted to investigate how the produced membranes’ surface charges changed in solutions with different levels of pH (3–12). A Zeta-Potential and Particle Size Analyzer (ELSZ-2000Z, Otsuka Electronics Co., Ltd., Osaka, Japan) was used to do the zeta potential analysis after a portion of the coating solution was dissolved in distilled water and the pH was adjusted to be between 3 and 12 units using HCL and NaOH.

### 2.6. TiO_2_/GO-Al_2_O_3_ HF NF Membrane Permeation Test, Mean Pore Size and Flux Recovery Rate

Epoxy resin glue was used to secure the TiO_2_/GO-Al_2_O_3_ HF NF membranes in the membrane housing (Figure 1). The experiment was run in a cross-flow filtration mode to minimize the impact of TiO_2_/GO-Al_2_O_3_ HF NF membrane contamination on the NF process. The pump’s operating pressure and flow rate were set at 5 bar and 50 mL/min, respectively. Every experiment was carried out at room temperature. To ensure stable operation of the NF membranes, TiO_2_/GO-Al_2_O_3_ HF NF membranes were pre-pressurized with distilled water at 7 bar for half hour prior to each experiment. Each experiment was carried out three times, with the results being averaged, to guarantee the repeatability of the data.

For pure water flow studies, distilled water was used as the feed, and once the membrane had been pre-pressured, the pressure was set for the experiments at 1, 2, 3, 4, and 5 bar. The permeated distilled water was collected for one hour after the pressure had stabilized, at which point the permeate volume was collected as *V*. The following formula was used to determine the permeate flux (*PF*):(3)$$PF=\frac{V}{A\cdot \Delta t}$$
where *A* and Δ*t* are the shell area of the TiO_2_/GO-Al_2_O_3_ HF NF membrane and the operation time, respectively.

For the filtration experiments, a feed salt solution concentration of 2000 ppm was used. The salt concentrations were measured by a conductivity measurement meter (Conductivity-ID944, IT Caster Ltd., China). The following equation was used to determine the salts’ retention rate:(4)$$R\left(\%\right)=\left(1-\frac{C_{P}}{C_{F}}\right)*100\;$$
where *C_P_* and *C_F_* are the permeate and feed concentrations, respectively.

PEG feed solutions with various molecular weights were utilized. Equation (3) could be used to calculate PEG retention rates. A total organic carbon analyzer (TOC-L, Shimadzu, Japan) was used to determine the PEG content. The molecular weight cut-off (MWCO) is typically defined as the molecular weight at 90% retention. Based on the average molecular weight, the Stokes diameter of PEG is calculated by the following equation:(5)$$d=0.0262\times M_{p}^{0.5}-0.03$$
where *d* is the diameter (nm) of the membrane and *M_p_* is the molecular weight (Da) of PEG.

The three-step membrane contamination test was conducted using the identical apparatus that is seen in Figure 1 and it proceeded as follows: (1) One pure water filtration used distilled water as a feed solution, and the membrane water flux was recorded as *J*_0_. (2) The lignin waste solution was utilized as a feed solution for filtration in the second part of the experiment. The membrane was then removed, and its surface was cleaned with a 0.1 mol/L NaOH solution before being washed with distilled water. (3) In the third step, the cleaned membrane was utilized again with pure water and then the water flux was recorded as *J*_1_. The transmembrane pressure used throughout the experiment was 5 bar, where the pure water flux was calculated using the following equation:(6)J=VA×∆t×P
where *V* is the volume of the permeate, *A* is the area, Δ*t* is the operation time and *P* is the transport pressure. In the end, the flux recovery ratio (*FRR*) was calculated using Equation (6), as follows:(7)FRR=J1J2×100

### 2.7. Long-Term Test in Lignin Wastewater

The stability tests for lignin and sodium ion rejection under long-term working conditions were also carried out using the apparatus shown in Figure 1. The permeate samples were added back to the feed solution after each analysis to avoid the effects of differential concentration polarization and to guarantee that the sodium ion and lignin concentrations in the feed solution do not vary. Another important consideration is the constancy of the water flux while the membrane is in operation so that the membrane throughput is calculated every 12 h.

## 3. Results and Discussion

### 3.1. Characterization of Al_2_O_3_ HF Support and TiO_2_/GO-Al_2_O_3_ HF NF Membrane

Figure 2 shows SEM photographs of a typical Al_2_O_3_ HF support and TiO_2_/GO-Al_2_O_3_ HF NF membranes. The surface morphologies of the Al_2_O_3_ HF support and TiO_2_/GO-Al_2_O_3_ HF NF membranes sintered at 1300 °C are shown in Figure 2a and Figure 2c, respectively.

The shell side surface of the fabricated Al_2_O_3_ HF support was defect-free and dense (Figure 2a). Al_2_O_3_ HF supports resulted in a thickness of about 310 µm (Figure 2b). In addition, the Al_2_O_3_ HF support cross-section (Figure 2b) illustrates a sponge-like structure in the middle area with finger-like structures near the lumen and shell sides. These macro voids with lengths up to approximately 110 μm were observed in the cross-section of the Al_2_O_3_ HF support at the lumen and shell sides (Figure 2b). The polymer phase inversion process is believed to be responsible for the formation of these structures [31,32,33,34]. The Al_2_O_3_ HF support produced in this study was covered with the TiO_2_ and GO layer (Figure 2c). The layer region of the TiO_2_/GO-Al_2_O_3_ HF NF membranes (Figure 2d) showed a TiO_2_ and GO layer with a thickness of up to 8 μm.

To confirm the nature of the deposited 8 µm-thick TiO_2_ and GO layer on the external surface of the Al_2_O_3_ HF support, we performed an EDX line and map analysis of the TiO_2_/GO-Al_2_O_3_ HF NF membrane interface. The colored lines and images highlight the distributions of the Al, Ti, and C elements. Figure 3b shows EDS mapping images of the TiO_2_/GO-Al_2_O_3_ HF NF membrane interface. The Ti and C elements were uniformly distributed in the deposited layer, which indicated that the TiO_2_ and GO composite layer can be co-deposited directly onto a porous Al_2_O_3_ HF support. Compared with the C, no Ti element distribution was present in the Al_2_O_3_ HF support, indicating that Ti could not be migrated to the bulk structures of the support. The Al element distribution was very uniform and limited to the Al_2_O_3_ HF support surface, indicating that there is no interdiffusion phenomenon between Al and Ti. A small amount of carbon should have been left over from the process of preparing the Al_2_O_3_ HF support body because the C element’s peak reflects the peak of Al. In conclusion, it is reasonable to suppose that the Al_2_O_3_ HF support was properly coated with the aforementioned composite layer.

The TiO_2_ and GO layer coated on the Al_2_O_3_ HF support can also be confirmed from the XRD analysis. Figure 4a shows a diffractogram of the Al_2_O_3_ HF support crystal structure and the results of the XRD analysis agree with the JCPDS data (Al_2_O_3_ Card no 10-0173). The XRD pattern of the Al_2_O_3_ HF support shows the main peak at 43.362°, indexed to the (113) plane of typical α-Al_2_O_3_ oxide (Corundum) and crystallizes in the hexagonal R-3c (No. 167) space group. The peak at low-angle at 25.584° corresponds to the (012) plane. More than 75% of the overall diffraction intensity over the chosen two theta ranges (20–80°) is represented by this peak. In addition, Figure 4a also shows that the TiO_2_ and GO layer coating process did not result in structural changes to the Al_2_O_3_ support.

Appendix A (Supplementary Material) shows the UV–Vis absorption spectra of the prepared GO. From Appendix A, it can be seen that the absorbance result for GO showed a peak at 241 nm. The absorption band centered at 241 nm is attributed to s to π–π* transitions of the remaining sp^2^ C=C bonds. These results are in good agreement with previous literature [35,36,37,38] and support the validity of the Hummers method to produce the GO powder [29,30].

As seen in Figure 4b, the crystal peak of the TiO_2_ and GO layers are well represented (TiO_2_, JCPDS Card No. 21-1272; Graphite, JCPDS Card No. 26-1079). In the diffractogram of the TiO_2_/GO-Al_2_O_3_ HF NF membrane, besides the diffraction peaks of Al_2_O_3_, there are several notable diffraction peaks at 25.3°, 48.08°, 53.89°, 55.07°, 32.688°, and 68.7°. These diffraction peaks can be well indexed to the (101), (200), (105), 211, 204, and (116) planes (JCPDS No. 21-1273), respectively, of the tetragonal phase of anatase (space group I41/amd (No. 141)). In contrast, a very weak diffraction peak at around 26.6°—corresponding to the (003) plane—is distinguishable for the rhombohedral graphite (JCPDS Card no. 26-1079) in the space group R3 (No. 146) [29,39]. Because the detection limit of the majority of XRDs is up to 5% by weight [40,41], trace levels of GO at a more high angle might have gone undetected. These findings validate the presence of the TiO_2_ and GO layer that was deposited on the Al_2_O_3_ HF support.

The successful co-deposition of TiO_2_ and GO layer on Al_2_O_3_ HF support is also reflected in the TGA curves. Figure 5 displays the TG curves of the Al_2_O_3_ HF support and TiO_2_/GO-Al_2_O_3_ HF NF membrane. From the TGA curve in Figure 5, it can be seen that there is a small weight loss difference between the Al_2_O_3_ HF support and the TiO_2_/GO-Al_2_O_3_ HF NF membrane where the gain in weight did not exceed 0.1%. The first step occurs at 25–75 °C, which may be due to the water molecule loss. The second step occurs at 170–250 °C, which may be due to the loss of labile oxygen groups (i.e., carboxylate, anhydride, lactone groups, etc.) [42]. The inset of Figure 5, which depicts the thermal events at around 70 °C and 170 °C, displays the TG curves of these samples between room temperature and 300 °C.

### 3.2. Surface Hydrophilicity and Porosity

In comparison to the Al_2_O_3_ HF support, Figure 6 illustrates the variation in the porosity (%) and water contact angle (%)—a quantitative measure of macroscopic surface wettability—for the TiO_2_/GO-Al_2_O_3_ HF NF membrane. After the TiO_2_ and GO coating process, the porosity of the TiO_2_/GO-Al_2_O_3_ HF NF membrane decreased from 54.3% of the Al_2_O_3_ HF support to 42.5%. In the meantime, the water contact angle increased from 26.89° to 35.43° with a TiO_2_ and GO layer on the Al_2_O_3_ HF support. Combining these findings also reveals that, although the Al_2_O_3_ HF support’s hydrophilicity decreased following the TiO_2_ and GO coating procedure, the reduction was not very significant and the high level of hydrophilicity of the TiO_2_/GO-Al_2_O_3_ HF NF membrane was still maintained.

The results of the variation in water contact angle with the time of the Al_2_O_3_ HF support and TiO_2_/GO-Al_2_O_3_ HF NF membrane are shown in Figure 7. The water contact angle of the Al_2_O_3_ HF support decreased in a non-continuous manner, from 26.9° when the experiment started to zero degrees after just 25 s. In the initial seconds of the test, the decrease in water contact angle was more evident. Instead, the water contact angle of the TiO_2_/GO-Al_2_O_3_ HF NF membrane decreased linearly from 35.43° to zero degrees after 35 s. Although the initial water contact angle increased as hydrophilicity decreased from the Al_2_O_3_ HF support to the TiO_2_/GO-Al_2_O_3_ HF NF membrane, the slope of the two results over time is quite similar, indicating that the coating simply influences the membrane’s hydrophilicity.

The surface zeta potential of the membrane is negative in both acid and alkaline solutions, indicating that a negative charge is fixed on the surface of the membrane (Figure 8). Therefore, the membranes are subject to electrostatic repulsion during filtration. The surface potential tends to rise when the pH is greater than about 10 pH units. The retention of ions is somewhat impacted by the electrical repulsion that the membrane demonstrates during separation.

Regarding the water permeate flux (L/m^2^h) versus pressure experiments of TiO_2_/GO-Al_2_O_3_ HF NF membrane (Figure 9), the experiment was repeated three times to evaluate the error of the water permeate flux. In all selected TiO_2_/GO-Al_2_O_3_ HF NF membranes, water permeate flux versus pressure could be fitted by using the linear function fitting: Y=A + Bx, where A equals the value of water permeate flux when the value of pressure is zero (x = 0) and B is the slope of the regression line. The fitting range was limited to the range between 1 and 6 bar. The obtained coefficients A and B and the resulting R are listed in Table 1. As can be noted in Table 1, the confidence (R) was determined to be >0.999 in all experiments.

The proposed TiO_2_/GO-Al_2_O_3_ HF NF membrane can be employed in a high-pressure environment since it has a three-point flexural strength of 28.97 Mpa (Table 2). In addition, the membrane’s FRR is 86.02%, which represents good performance in terms of pollution resistance. This means that, after a specific amount of time, the membrane can be rinsed and then utilized again.

### 3.3. Permeation Test

Since the surface of the TiO_2_/GO-Al_2_O_3_ HF NF membranes has electrical properties in an aqueous solution and according to the characteristics of dielectric repulsion, ions with higher valence will be retained due to dielectric repulsion, as can be seen from Figure 10, the retention rate of monovalent cations by the TiO_2_/GO-Al_2_O_3_ HF NF membranes is much lower than that of multivalent ions. The Donnan equilibrium effect can be utilized to explain the rejection function of the charged TiO_2_/GO-Al_2_O_3_ HF NF membranes [17]. The retention rate of SO_4_^2−^ ions by the membrane is lower than that of Cl^-^ due to the Donnan equilibrium effect and electrostatic repulsion effect, whereas the retention rate of bivalent ions (Ca^2+^, and Mg^2+^) is higher than that of monovalent ions (Na^+^, and K^+^). As can be seen in the inset of Figure 10. the retention rates are in line with the hydrated radius of each cation (Na^+^, Mg^2+^, K^+^, Ca^2+^, Mg^2+^, and Al^3+^) [43]. These results are in good agreement with values obtained using the γ-Al_2_O_3_ film-coated porous α-Al_2_O_3_ hollow fiber membrane [17,18].

According to Figure 11, the developed TiO_2_/GO-Al_2_O_3_ HF NF membranes have a cut-off of 89.11% for PEG molecules with a molecular weight of 600 Da and a cut-off of 93.11% for molecules with a molecular weight of 800 Da. The molecular weight cut-off of the investigated TiO_2_/GO-Al_2_O_3_ HF NF membranes is therefore estimated to be between 600 and 800 Da. The TiO_2_/GO-Al_2_O_3_ HF NF membrane shows the properties of an NF membrane with MWCO of approximately 600–800 Da, as shown in Figure 11.

### 3.4. Long-Term Separation Test

Over the course of a 120-h experiment, the flux, lignin rejection, and ion rejection of the TiO_2_/GO-Al_2_O_3_ HF NF membrane in lignin wastewater were regularly rerecorded. Figure 12 shows the results of the long-term separation test. Figure 12 reveals that the retained lignin in the lignin wastewater was retained by the TiO_2_/GO-Al_2_O_3_ HF NF membrane at a rate of 92% while the retained sodium ions were retained at a rate of 5.2%. This indicates that the studied TiO_2_/GO-Al_2_O_3_ HF NF membrane was effective in separating the lignin and sodium ions from the lignin wastewater.

Additionally, as shown in Figure 12, the membrane was able to sustain a water flux of more than 14.5 L/m^2^h throughout the prolonged operation in lignin wastewater. The results shown in Figure 12 show that there is a significant decrease in the flux of lignin after twelve hours of operation. The lignin in the lignin wastewater has a sticky nature that tends to adhere to the surface of the TiO_2_/GO-Al_2_O_3_ HF NF membrane, resulting in the pores in the membrane being blocked by the adhesion of lignin when the experiment was started. However, the membrane’s flow was still maintained at a good level after twelve hours. As a result, it is clear that the membrane does not diminish over the extended operation (5 days) and stabilizes at around 14.5 L/m^2^h.

## 4. Conclusions

This work effectively developed and utilized a new TiO_2_/GO-Al_2_O_3_ HF NF membrane for desalination and lignin recovery from lignin effluent. The novel TiO_2_/GO-Al_2_O_3_ HF NF membrane can achieve a sodium retention rate of more than 5.2% in lignin wastewater. In conclusion, the present TiO_2_/GO-Al_2_O_3_ HF NF membrane can achieve an excellent lignin retention rate of more than 92%. The TiO_2_/GO-Al_2_O_3_ HF NF membranes also are stable upon a prolonged lignin wastewater separation process.

## Figures and Tables

**Figure 1 membranes-12-00950-f001:**
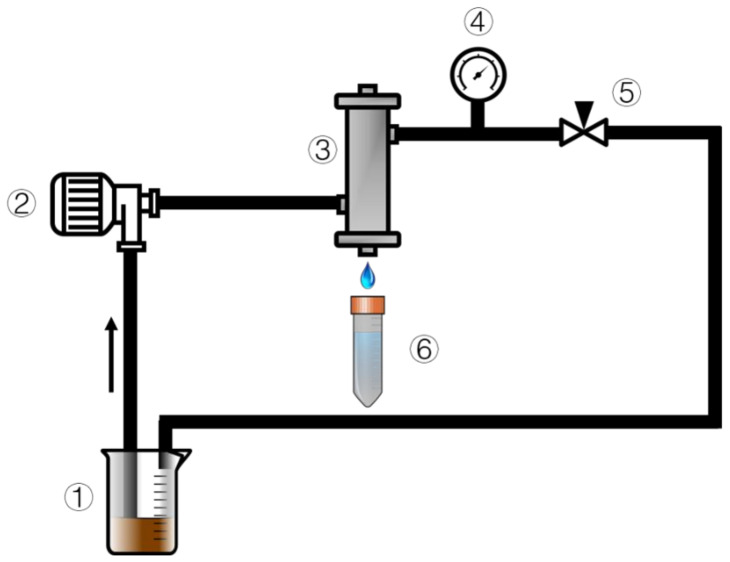
Diagram of the experimental setup of the cross-flow rate: 1 = feed, 2 = pump, 3 = membrane module, 4 = pressure meter, 5 = valve, and 6 = permeate.

**Figure 2 membranes-12-00950-f002:**
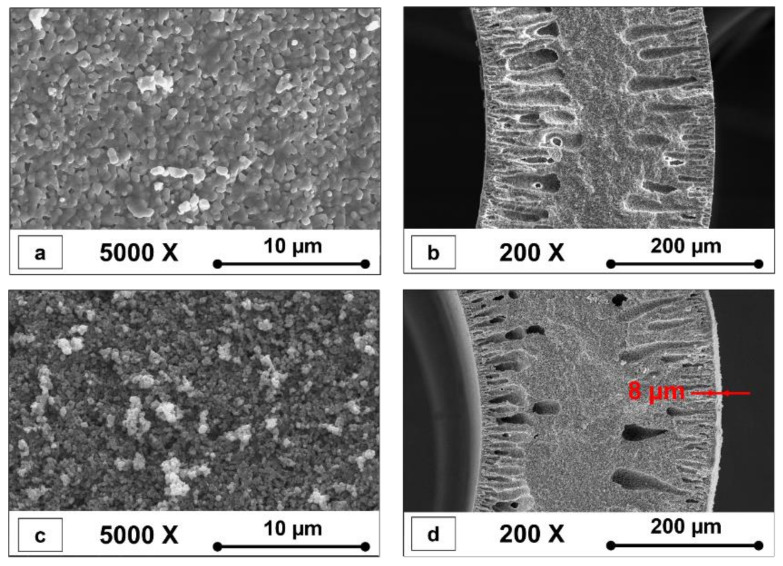
SEM images of (**a**) external surface and (**b**) cross-section of Al_2_O_3_ HF support, and (**c**) surface and (**d**) cross-section of TiO_2_/GO-Al_2_O_3_ HF NF membrane.

**Figure 3 membranes-12-00950-f003:**
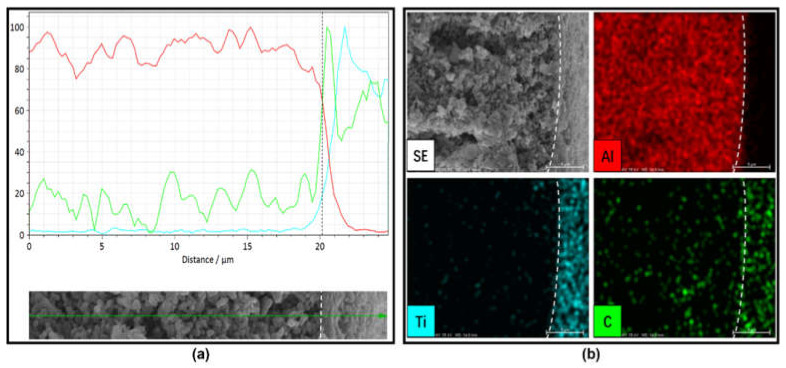
EDS (**a**) line analysis and (**b**) map analysis for the TiO_2_/GO-Al_2_O_3_ HF NF membrane interface. The distributions of the Al, Ti, and C elements are highlighted by lines and images using different colors.

**Figure 4 membranes-12-00950-f004:**
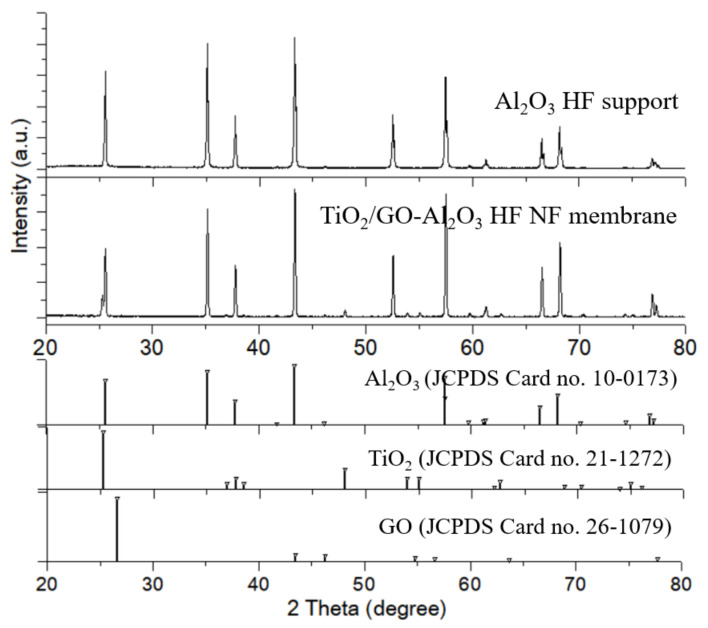
XRD of the Al_2_O_3_ HF support and TiO_2_/GO-Al_2_O_3_ HF NF membrane, along with standard XRD pattern of graphite for Al_2_O_3_ (JCPDS Card No. 10-0173), TiO_2_ (JCPDS Card No. 21-1272) and graphite (JCPDS Card no. 26-1079) for reference.

**Figure 5 membranes-12-00950-f005:**
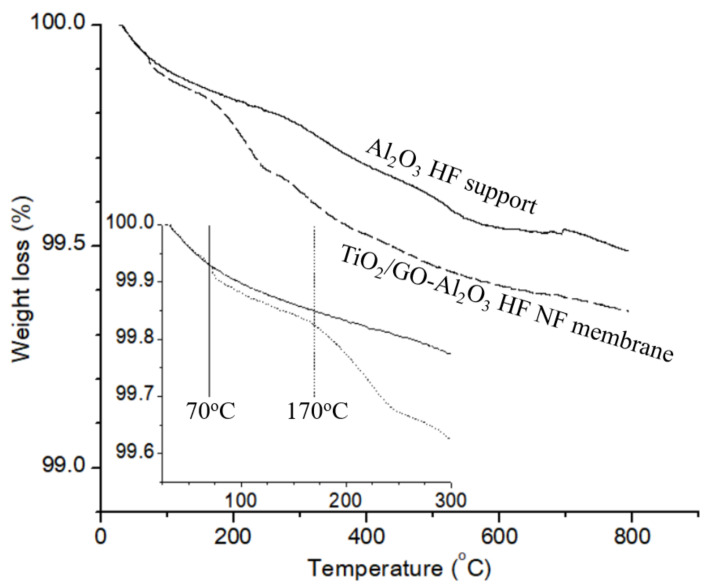
TG curves of the Al_2_O_3_ HF support and TiO_2_/GO-Al_2_O_3_ HF NF membrane. Inset: Initial weight loss (%) values (25–300 °C).

**Figure 6 membranes-12-00950-f006:**
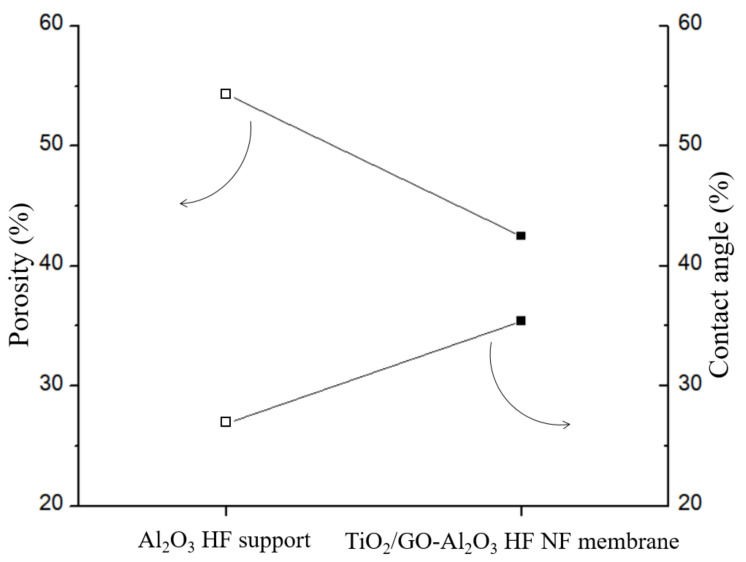
Porosity variation (%) and water contact angle variation (%) between Al_2_O_3_ HF support and TiO_2_/GO-Al_2_O_3_ HF NF membrane.

**Figure 7 membranes-12-00950-f007:**
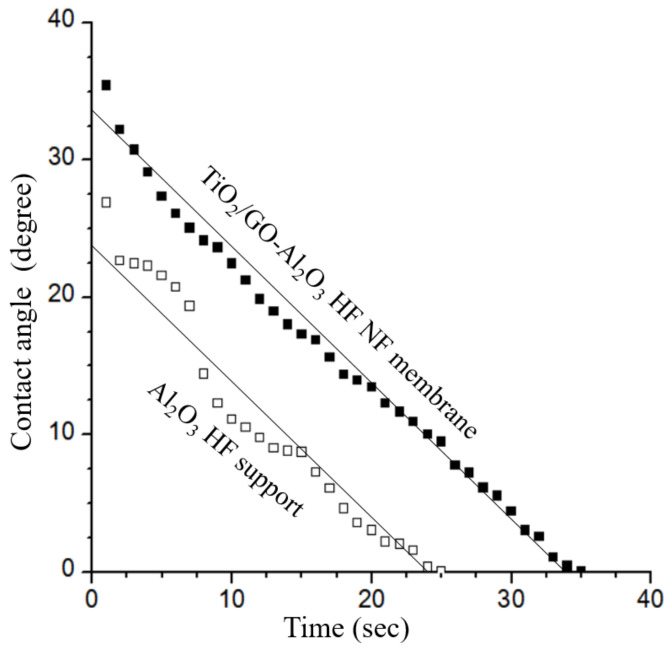
Water contact angle (°) versus time of the Al_2_O_3_ HF support and TiO_2_/GO-Al_2_O_3_ HF NF membrane.

**Figure 8 membranes-12-00950-f008:**
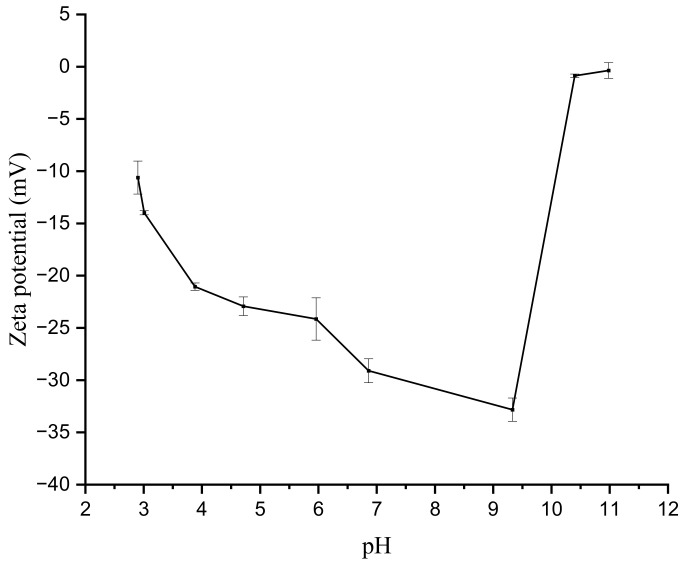
Surface zeta potential values of TiO_2_/GO-Al_2_O_3_ HF NF membranes at different pH values. Figure 9 shows the water permeate flux (L/m^2^h) versus pressure (bar) of the TiO_2_/GO-Al_2_O_3_ HF NF membrane at room temperature. The results of the water permeate flux experiments show that a pressure-driven NF process occurs in the studied TiO_2_/GO-Al_2_O_3_ HF NF membrane, as evidenced by the fact that water flux increases as operating pressure increases. The water flux at low pressure (i.e., 1 bar) is around 5.4 L/m^2^h, which is higher than the flux we previously achieved when we used an ultra-thin γ-Al_2_O_3_ film-coated porous α-Al_2_O_3_ hollow fiber (HF) support [17]. The water permeate flux increases from 5.4 to 33.3 L/m^2^h as the pressure increases from 1 to 6 bar.

**Figure 9 membranes-12-00950-f009:**
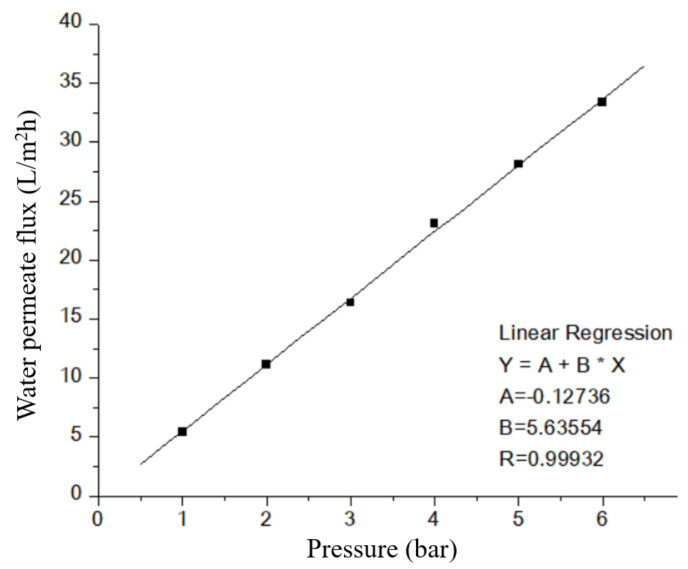
Water permeate flux (L/m^2^h) versus pressure of TiO_2_/GO-Al_2_O_3_ HF NF membrane. The line represents the linear fit to the experimental data.

**Figure 10 membranes-12-00950-f010:**
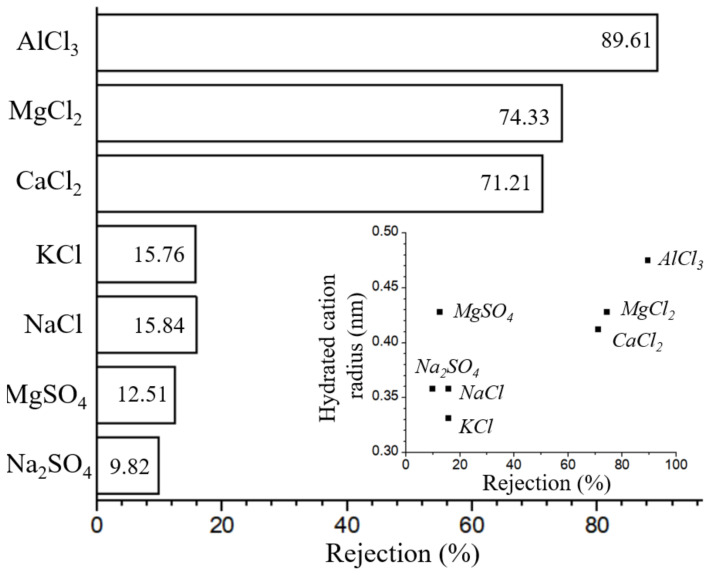
Na_2_SO_4_, MgSO_4_, NaCl, KCl, CaCl_2_, MgCl_2_, AlCl_3_ rejections of the TiO_2_/GO-Al_2_O_3_ HF NF membrane. Inset: obtained rejection (%) versus hydrated radius of cations (Na^+^, Mg^2+^, K^+^, Ca^2+^, Mg^2+^, and Al^3+^) [43].

**Figure 11 membranes-12-00950-f011:**
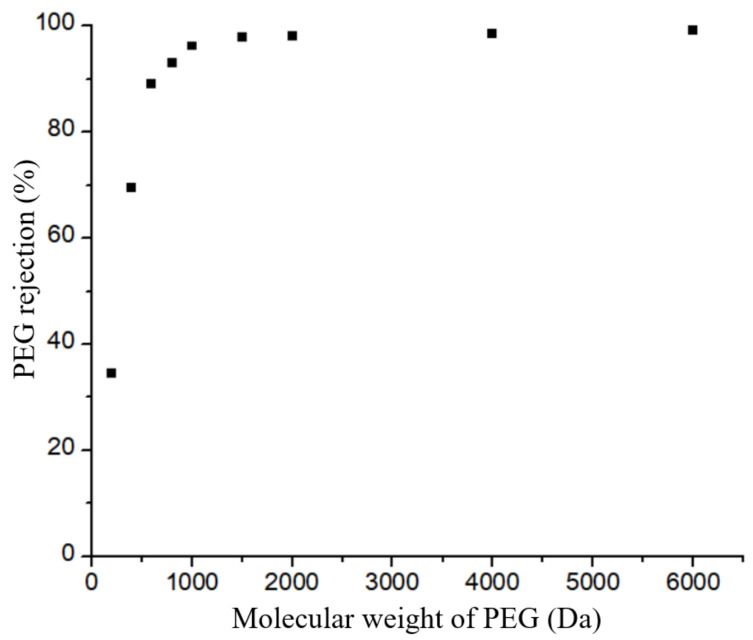
PEG rejection of the TiO_2_/GO-Al_2_O_3_ HF NF membrane.

**Figure 12 membranes-12-00950-f012:**
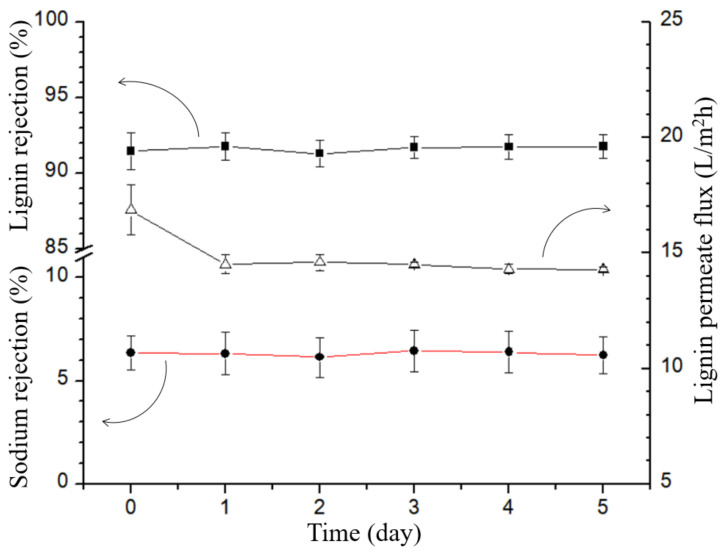
Retention of sodium ions, retention of lignin and lignin permeate flux in lignin wastewater. For the average statistics, each experiment was run three times in the same conditions.

**Table 1 membranes-12-00950-t001:** Fitting results of water contact angle (°) versus time using linear function for three Al_2_O_3_ HF supports and TiO_2_/GO-Al_2_O_3_ HF NF membranes.

Pressure (bar)	Water Permeate Flux (L/m^2^h)		
	1st	2nd	3rd
1	5.452677	5.619573	5.328018
2	11.16083	10.98572	11.1781
3	16.36402	16.43361	16.37582
4	23.1098	22.73496	23.19718
5	28.11514	27.69619	28.07752
6	33.37971	33.37116	33.29028
Fitting results			
A	−0.12736	−0.04553	−0.15861
B	5.63554	5.57688	5.63803
R	0.99932	0.9997	0.99914

**Table 2 membranes-12-00950-t002:** Flux recovery ratio (FRR, %) and three-point bending strength (MPa) of TiO_2_/GO-Al_2_O_3_ HF NF membranes.

Bending Strength (MPa)	Stand. Dev.	FRR (%)	Stand. Dev.
28.97	7.81	86.03	2.11

## Data Availability

Not applicable.

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
