# Peer review of "Novel TiO2/GO-Al2O3 Hollow Fiber Nanofiltration Membrane for Desalination and Lignin Recovery"

_membranes, 2022, doi:10.3390/membranes12100950_

Round 1

Reviewer 1 Report

Review of the article " Novel TiO2/GO-Al2O3 Hollow Fiber Nanofiltration Membrane for Desalination and Lignin Recovery".

The article "Novel TiO2/GO-Al2O3 Hollow Fiber Nanofiltration Membrane for Desalination and Lignin Recovery" is devoted to the development of new membranes on a hollow fiber substrate based on polyethersulfone and α-Al2O3 with a deposited selective layer of TiO2/GO. The membranes were developed for the first time and have shown high retention rates of lignin from water. At the same time, there are a number of questions about the work.

1)             Lines 33-34. For what "other" purposes can lignin be used?

2)             What is the supposed size of lignin?

3)             An increase in hydrophobicity is a negative factor for filtration membranes due to possible clogging of the membrane surface and a decrease in its performance. Have the flux recovery ratio (FRR) of pure water and the surface of membranes been studied by the SEM and IR methods after water filtration with lignin?

4)             Why is the retention coefficient 15.84% when Na + is separated from water (Fig. 9), and when Na + is filtered from a mixture with lignins, the retention coefficient is 5.2%  (Fig. 10)?

5)             It is necessary to show how these membranes are better than other nanofiltration polymer membranes. Indeed, the addition of α-Al2O3 to PES and the deposition of a TiO2/GO layer significantly increases the price of the membrane, despite the fact that it is not a panacea for separating lignin and, moreover, cations and anions of salts from water.

Reviewer 2 Report

This study investigates the preparation of TiO2/GO-Al2O3 hollow-fiber nanofiltration membrane for desalination and lignin recovery. In the study, the authors prepared and analyzed the properties of the synthesized membranes such as morphology (by SEM), structural properties (by XRD), hydrophilicity, and porosity. Then, the membrane permeability and MWCO were determined. The filtration efficiency was analyzed by comparing salts and lignin rejection as well as water flux during long-term filtration of lignin wastewater.

 In my opinion, the manuscript was good-written and organized. However, several points should be clarified, and improved the quality of the manuscript before accepting for publication.

Below, I present my comments, questions, and suggestions that may strengthen the scientific quality of the paper.

1.           Introduction

Overall comments:

- The novelty of the article should be more strongly accented. Moreover, the Authors should describe the NF process in more detail and justify the choice of this separation technique as a suitable method of lignin removal.

-             Line 53 - Please rewrite this sentence (Repeating of polyethersulfone).

2.1 Materials

- Line 86 - Please consistently use the abbreviation PES or PESf for polyethersulfone.

- Line 103 - Add a space after the unit (3 Mpa).

2.4 Preparation of TiO2/GO-Al2O3 hollow fiber (HF) nanofiltration (NF) membrane

- Line 130 - Please change GC to GO.

2.6 TiO2/GO-Al2O3 HF NF membrane permeation test and mean pore size

- What is the maximum working TMP range for synthesized membranes?

- How was the lignin content/concentration in the feed solution determined?

3.1 Characterization of Al2O3 HF support and TiO2/GO-Al2O3 HF NF membrane

- In my opinion, the entire manuscript lacks significant information about the zeta potential of the synthesized membrane. As is well known, the surface potential of the NF active layer has a significant impact on solute rejection. Therefore, the authors should provide information about the membrane's zeta potential over a wide pH range.

- Fig. 3 and 4 - Please add the markings a), b), c), etc.

3.4. Long-term separation test

-From a practical point of view, the authors should provide information on an effective membrane cleaning procedure after the NF process of lignin wastewater. In addition, the authors should indicate the information about the lifetime of the synthesized membranes. In my opinion, the long test may be repeated in several cycles: cleaning-separation.

- Please provide information on how many repetitions of the experiment were made. The error bars are missing in Fig. 11.

Round 2

Reviewer 2 Report

Thank you for completing the manuscript. Accepts the manuscript as it stands.